# Unveiling the Fundamental Mechanisms of Graphene Oxide Selectivity on the Ascorbic Acid, Dopamine, and Uric Acid by Density Functional Theory Calculations and Charge Population Analysis

**DOI:** 10.3390/s21082773

**Published:** 2021-04-14

**Authors:** Kittiya Prasert, Thana Sutthibutpong

**Affiliations:** 1Theoretical and Computational Physics Group, Department of Physics, Faculty of Science, King Mongkut’s University of Technology Thonburi (KMUTT), Bangkok 10140, Thailand; kittiya.sert@mail.kmutt.ac.th; 2Center of Excellence in Theoretical and Computational Science (TaCS-CoE), Faculty of Science, King Mongkut’s University of Technology Thonburi (KMUTT), 126 Pracha Uthit Rd., Bang Mod, Thung Khru, Bangkok 10140, Thailand

**Keywords:** density functional theory, AA, UA, and DA detections, graphene oxide

## Abstract

The selectivity of electrochemical sensors to ascorbic acid (AA), dopamine (DA), and uric acid (UA) remains an open challenge in the field of biosensing. In this study, the selective mechanisms for detecting AA, DA, and UA molecules on the graphene and graphene oxide substrates were illustrated through the charge population analysis from the density functional theory (DFT) calculation results. Our substrate models contained the 1:10 oxygen per carbon ratio of reduced graphene oxide, and the functionalized configurations were selected according to the formation energy. Geometry optimizations were performed for the AA, DA, and UA on the pristine graphene, epoxy-functionalized graphene, and hydroxyl-functionalized graphene at the DFT level with vdW-DF2 corrections. From the calculations, AA was bound to both epoxy and hydroxyl-functionalized GO with relatively low adsorption energy, while DA was adsorbed stronger to the electronegative epoxy groups. The strongest adsorption of UA to both functional groups corresponded to the largest amount of electron transfer through the pi orbitals. Local electron loss created local electric fields that opposed the electron transfer during an oxidation reaction. Our analysis agreed with the results from previous experimental studies and provided insight into other electrode modifications for electrochemical sensing.

## 1. Introduction

One of the open problems in the field of biosensing is the simultaneous detection of ascorbic acid (AA), dopamine (DA), and uric acid (UA) within body fluids. The lack of ascorbic acid (AA) can cause scurvy and other diseases [1], while an abnormal level of the dopamine (DA) neurotransmitter is related to mental disease conditions [2], and the level of uric acid (UA) can identify the symptom for gout [3]. AA, DA, and UA are oxidizable, so electrochemical biosensors can detect the amount of these biomarkers up to the detection limit of the micromolar range. However, AA, DA, and UA molecules possess almost similar oxidation potentials, resulting in strong interference and overlapping responses in cyclic voltammetry (CV) and differential pulse voltammetry (DPV) experiments with conventional glassy carbon electrodes. Moreover, the concentration of DA in human serum is linked to the mental status of the patient [4], which might further cause uncertainty and false-positive detections of AA and UA [5]. To overcome this problem from the CV peak overlapping, modification of the electrode is necessary.

Carbon-based 2D materials, e.g., graphene [6], graphene oxide (GO), and reduced graphene oxide (rGO), have been widely used in electrochemistry due to their low cost, wide potential window, and being relatively inert to electrochemical reaction [7]. Graphene possesses many interesting properties, including high charge mobility, excellent electrical and thermal conduction, high mechanical strength, perfect biocompatibility, and low toxicity [8]. Moreover, the surface of graphene can be adjusted by introducing defects or chemical functionalization [9]. These made graphene an ideal material for developing biosensors. Many studies have reported electrode modification to improve the simultaneous determination of AA, DA, and UA in the CV and differential pulse voltammetry (DPV) experiments [10,11,12,13,14,15,16,17,18,19,20,21,22], e.g., the use of multilayer graphene nanoflake films (MGNFs) [10], graphitic sheets or multi-walled carbon nanotube/metal composites [11,13,14,17,18,19,20,21], and conducting polymers [12,15,16,22]. Moreover, electrical [23] and chemical [24] reduced graphene oxides (rGO) were also employed for AA/DA/UA detections due to the better conductivity than GO and their cost-effectiveness. In recent studies, carbon-based electrodes were also modified by MnO_2_, carbon nitride nanosheets (C_3_N_4_), polypyrrole, or ionic liquids that could extensively alter the redox potentials [25,26,27,28]. Those previous studies proposed that metals decreased the reduction potential. Meanwhile, carbon-based materials accelerated the oxidation and amplified the oxidation peak separation, and the porous surface exhibited the thin layer character that facilitated the discrimination of molecules with overlapping redox potentials. However, discussions on the relationships between the intrinsic molecular properties of the analytes and the selectivity of the sensors are still lacking.

In this study, relatively simplified model systems were considered to address the fundamental mechanisms of the oxidation potential shift of AA, DA, and DA molecules. The binding of these analytes to graphene oxide models was elucidated through a series of density functional theory (DFT) calculations. Firstly, representative configurations of GO were selected by the lowest formation energy. Then, geometry optimizations were performed on AA, DA, and DA analyte molecules on pristine graphene, a graphene model functionalized by hydroxyl groups, and a graphene model functionalized by epoxy groups. After that, partial charge analysis was performed on the AA, DA, and DA analytes in cases for free analytes, oxidized forms, and the molecules bound to different substrates. Functionalization of a pristine graphene plane by epoxy and hydroxyl groups could result in the altered charge distribution patterns of the analyte and might affect the tendency of oxidation reactions.

## 2. Computational Methods

All calculations in this study were based on density functional theory (DFT), using the Quantum Espresso 5.3 package [29]. For the exchange-correlation of electrons, GGA-PBE functional [30,31] was employed, while the projector augmented wave method was implemented for the electrons–nucleus interactions. Electron kinetic energy cut-off for convergence of plane-wave expansion was set to 50 Ry for all calculations, and the van der Waals interaction for long-range electrons correlation was corrected by the vdW-DF2 approach of the Thonhauser group [32,33]. The first Brillouin zones sampling of supercell was determined by 11 × 11 × 1 Monkhorst-Pack grids [34]. The convergence of SCF iteration loops was set with the energy tolerance 1 × 10^−6^ Ry, and the convergence of geometry optimization loops was set with the force tolerance of 0.001 Ry/Å.

Monolayer pristine graphene and graphene oxide supercells of the dimension 5 × 5 were created. The vertical slab distance was set up at 20 Å to prevent surface interaction from the above graphene layer under the periodic boundary condition. The formation energy of *GO* surface containing n epoxy or n hydroxyl groups, $E_{form}$ was calculated by
(1)$$E_{form}=E_{GO}-E_{G}-nE_{O/OH},$$
where $E_{GO}$, $E_{G}$, and $E_{O/OH}$ are the total energy of the *GO* surface, the pristine graphene substrate, and a free epoxy (*O* atom) or a free hydroxyl (*OH*) group. Formation energy is the energy required for each structure to form a configuration. In order to estimate the most energy favorable of surface structures, $E_{form}$ values are normalized to define the formation energy per functional group as
(2)$$E_{norm}=\frac{E_{form}}{n},$$

Then, the optimized structure of each analyte was translated along the vertical axis (*z*-axis) so that the center of the ring was at distances 2.0 Å to 8.0 Å from the substrate, and the potential energy was calculated as a function of vertical distance. The potential energy E(r) was then fit to the Morse potential function:(3)$$E\left(r\right)=E_{0}+D_{e}\left(e^{-2a\left(r-r_{0}\right)}-2e^{-a\left(r-r_{0}\right)}\right),$$
when $E_{0}$ was the offset of the minimum energy. The adsorption energy was defined from the well depth ($D_{e}$) parameter, while the equilibrium binding distance can be obtained from the parameter ($r_{e}$). Additionally, the binding stiffness about the equilibrium ($k_{e}$) could be obtained from $k_{e}=2D_{e}\alpha^{2}$.

All optimized configurations of analytes binding on all substrates were visualized through the VESTA software [35]. A 0.005 e/Å^3^ isosurface of charge density was created for each configuration to estimate the van der Waals radius, and the ±0.001 e/Å^3^ isosurfaces were created to visualize the difference between the charge density of an analyte before and after substrate binding. The partial charge was calculated for each atom from the Löwdin population [36] to analyze the differences between partial charges of a neutral analyte molecule and the analyte (i) in oxidized form, (ii) binding with pristine graphene, (iii) binding with GO5-para, and (iv) binding with G(OH)5-paraA.

## 3. Results

### 3.1. Graphene Oxide Configuration

From an experimental perspective, oxygen functionalization on graphene oxide can occur in many different configurations [37,38]. Studies by Lerf and Klinovski’s group [39,40] confirmed that epoxy and hydroxyl groups are the major functional groups on graphene oxide. Based on the experimental results, the Lerf-Klinovski model suggested that epoxy and hydroxyl groups on the graphene oxide surface were aggregated as clusters with some hydroxyl, carbonyl, and carboxyl groups at the edge. As a consequence, non-oxidized areas or aromatic islands were also found on graphene oxide. Similar trends were observed from previous DFT studies that the GO structures with clustered functional groups were more stable than the GO structures with isolated functional groups [41,42]. In this work, graphene supercells of dimensions 5 × 5 were functionalized with five hydroxyl or epoxy functional groups using the configurations proposed by Domancich et al. [42] (Figure 1). This degree of functionalization was selected due to the highest stability measured by the formation energy per functional group. Additionally, the 10:1 carbon to oxygen ratio of GO models in this study resembled the ratio of reduced-GO [43,44,45,46], which could simultaneously detect AA, DA, and UA at separated oxidation voltages. After geometry optimization, formation energy ($E_{form}$) for each functionalized configuration was calculated to measure the conformational stability. Table 1 showed that functionalization of both epoxy and hydroxyl groups onto the graphene surface, using configurations proposed by Domancich et al. [42], were exothermic processes. Formation energy per one epoxy group was found between −3.092 and −3.231 eV, larger than the formation energy per one hydroxyl group, between −1.191 and −1.440 eV.

Relative stability among the four epoxy-functionalized GO configurations (Figure 1a–d) was determined from comparing the $E_{form}$ per functional group in Table 1. Our DFT calculation using the GGA-PBE functional with vdW-DF2 correction found that the stability of GO5-zigzag-B > GO5-para > GO5-zigzag-A > GO5-armchair was with a similar trend to the previous DFT study [20]. The most stable epoxy-functionalized GO, GO5-zigzag-B, was with an epoxy group at the opposite side of the graphene plane from the other four epoxy groups. The minimized distortion Δz of GO5-zigzag-B reduced the stress of the graphene plane. This behavior was confirmed by the previous DFT work by Yan et al. [15]. The second most stable configuration was GO5-para, where all the pairs of carbon atoms attached to all epoxy groups were aligned in parallel, and all ten occupied carbon atoms formed a cluster. Even though the epoxy functionalization GO5-para caused the largest plane distortion (highest Δz), the curvature of the graphene surface introduced by epoxy groups would facilitate further functionalization as less energy penalty was required for sp2 to sp3 transition when the sp2 orbitals were distorted out of the graphene plane. For applications on molecular sensors, GO and rGO could be in multilayer forms and only one side of the surface was interested. Therefore, GO5-para was picked for further investigation on the AA, DA, and UA interactions.

Relative stability from the $E_{form}$ per functional group among the four hydroxyl-functionalized GO configurations was found as ordered by G(OH)5-paraA > G(OH)5-paraB > G(OH)5-paraD > G(OH)5-paraC (Figure 1e–h and Table 1). Similar to the GO5-zigzag-B configuration, the G(OH)5-paraD configuration contained a hydroxyl group bound to the opposite side of the graphene plane from the other four bound hydroxyl groups. The G(OH)5-paraD configuration had the smallest surface distortion compared to other hydroxyl-functionalized GO configurations and was with the second-lowest formation energy. The G(OH)-paraA model configuration provided the lowest formation energy, corresponding to the smallest clusters of functional groups forming three hydrogen bonds, and was selected for further investigation on the AA, DA, and UA interactions.

### 3.2. Charge Distribution and the Potential Surface of AA/UA/DA

DFT calculations were performed for isolated AA, DA, and UA to understand the molecular basis of the interactions between the proposed graphene oxide surface with AA, DA, and UA analyte molecules. After each DFT calculation, Löwdin charge population on each atom, electron density isosurface, and electrostatic potential were extracted. Figure 2 displays the structures and the atomic nomenclatures of neutral AA, DA, and UA molecules. For each analyte molecule, atom groups were defined by the dashed circles according to ring members and their adjacent atoms. Partial charge on each atom from the Löwdin population analysis, along with the summation of partial charges for all the atom groups, were summarized in Table 2. Local polarity and contribution to the electrostatic potential for substrate binding were discussed in terms of atomic and group partial charges. Moreover, changes in charge distribution among the ring members would be discussed when the analyte was oxidized or bound to the substrate. For each molecule, the atom group with the highest positive charge was denoted by the ‘***’ sign, and the ‘Ox’ sign denoted the oxidation site. Figure 2a displayed the atomic nomenclatures of all constituent atoms and atom groups of a neutral AA molecule. The high potential region within the furanose ring was contributed by the polarity of the C1/O6/H8, C3/O2, and C4/O3/H1 groups, in which the ring carbon atoms were positively charged. However, the highest positive group partial charge of +0.211 e was found at the C2/H2 group near the most negatively charged O4 group with the absence of neither hydroxyl nor carbonyl groups, causing the higher electrostatic potential relative to other regions (represented by the blue color) and a higher affinity to bind with the negatively charged epoxy functional group of graphene oxide. Figure 2b displayed the atomic nomenclatures of all constituent atoms and atom groups of a neutral DA molecule. Like AA, the positive potential region at the center of the phenyl ring was also caused by the polarity of the ring carbon atoms covalently bonded with the outer oxygen atoms. The highest positive group partial charge of +0.098 e was found at the C7/O2/H11 group, also serving as one of the oxidation sites for DA. Figure 2c displayed the atomic nomenclatures of all constituent atoms and atom groups of a neutral UA molecule. The positive potential regions at the center of both 5-membered and 6-membered rings were caused by the relatively strong dipole-moments of all carbonyl (C=O) groups. The strong polarity of carbonyl groups and the electronegativity of nitrogen atoms N2 and N3 contributed to the highest positive partial charge of +0.219 e for the C3 atom adjacent to the H2/N2 oxidation site.

Now, consider the electrostatic potential of AA, DA, and UA molecules mapped onto the 0.005 e/Å^3^ iso-density surfaces in the right panels of Figure 2, approximated as the van der Waals surfaces of the molecules. Highly positive electrostatic potential on the surfaces represented in blue color indicated the preferred sites for the analyte molecules on the negatively charged functional groups of GO surfaces. The high potential regions on the surface of AA were both found within (labeled as 1°) and outside (labeled as 2°) the furanose ring, close to the most positively charged C2/H2 group (Figure 2a right). The secondary high potential region outside the planar structure suggested an additional off-plane binding site for AA. Similar to AA, two high potential regions were found within (labeled as 1°) and outside (labeled as 2°) the phenyl ring between the oxidation sites (Figure 2b right). However, the primary positive potential was relatively higher than the secondary positive potential, suggesting that the negatively charged functional groups of GO should mostly adsorb DA molecules through the phenyl ring of DA. For the UA molecule (Figure 2c right), the primary (labeled as 1°) and the secondary (labeled as 2°) high potential regions were located at the 6-membered and the 5-membered rings, respectively. Both high potential regions served as the preferred binding site on the functional groups of GO, suggesting that the aromatic rings contributed to the whole adsorption between the UA molecule and the GO surface.

### 3.3. Binding Configurations and Binding Strengths of AA/UA/DA on Graphene and GOs

From a previous DFT study on the neutral DA adsorption on a pristine graphene surface, parallel orientations of the aromatic ring of DA in both AA and AB configurations were the most energetically favorable [47]. Therefore, the parallel orientation of AA, DA, or UA on pristine graphene, GO5-para, or G(OH)5-paraA was proposed in all starting configurations in this study. For the pristine graphene substrate, an analyte molecule was placed near the center of the supercell with each of the ring atoms oriented most directly on top of a carbon atom in hexagonal lattice before the geometry optimization to maximize pi-pi stacking interactions. Meanwhile, for the case of binding on the GO5-para, or G(OH)5-paraA substrates, AA, DA, or UA were placed so that the analyte molecules covered most of the functional groups of graphene oxides to maximize the amount of van der Waals contacts. Optimized binding configurations of all three analyte molecules on all three substrate models were displayed in Figure 3, along with the 0.005 e/Å^3^ iso-density surfaces that roughly represent the van der Waals surfaces. Then, the optimized structure of each analyte was translated along the vertical axis (*z*-axis), and the potential energy was calculated as a function of vertical distance. The potential energy curve was then fit to the Morse potential function, and the adsorption energy can be reproduced from the well depth ($D_{e}$) parameter, while the equilibrium binding distance can be obtained from the parameter ($r_{e}$). Additionally, the binding stiffness about the equilibrium ($k_{e}$) could be obtained from $k_{e}=2D_{e}\alpha^{2}$ (Table 3).

Figure 3a–c displayed the optimized binding configuration of AA, UA, and DA on a 5 × 5 pristine graphene supercell. All three analyte molecules maintained their parallel orientation on the graphene plane. Slight horizontal shifting was seen for AA and DA to avoid steric clashes between their off-plane hydrogen atoms and graphene surfaces, while almost no shifting was seen for the planar UA molecule. Dipole and quadrupole moments from the non-uniform distribution of electrons induced a non-uniform electron distribution on the graphene plane and caused an additional weak electrostatic attraction. While maintained their horizontal orientation, high-density regions were found between all analyte molecules and the graphene plane. High-density regions were found at the off-plane hydrogen atoms with the positive partial charge of AA (atom H2) and DA (atoms H7 and H8), causing the higher electron density within a region of the graphene plane below those atoms. For UA, a small region between the analyte and the graphene plane with high electron density was found around the C3 atom with the highest partial charge. From the analysis of binding potential energy between the analyte and the substrate, the largest equilibrium distance $r_{e}$ was found at 3.573 Å for the DA molecule. UA was with both the highest energy and stiffness for adsorption on the pristine graphene as the molecule was purely planar and cyclic, allowing the largest number of pi-pi stacking with graphene. The analyte of the second-highest adsorption energy and stiffness was DA, with its higher aromaticity than that of AA.

Figure 3d–f showed the optimized binding configuration of AA, UA, and DA on the GO5-para substrate. The analyte molecules were slightly reoriented so that the high potential regions became closer to the electronegative oxygen atoms of the epoxy groups. Figure 3d showed that the C2/H2 atom group (denoted by ‘***’) of AA within the primary high potential region was in close contact at a distance of 3.052 Å from the epoxy group at the middle. The curvature of epoxidized GO substrate corresponded to the absence of the surface contact between the other epoxy groups and the secondary high potential region. Compared with the pristine graphene binding, the adsorption energy of AA was decreased to 0.031 Ry, and the binding stiffness was decreased to 0.111 Ry/Å^2^. For the case of DA molecule binding with the epoxidized GO in Figure 3e, the primary high potential region of DA at the middle of the phenyl ring and the secondary high potential region between two hydroxyl groups of DA were bound to three epoxy groups of the epoxidized GO. The DA molecule was reoriented on the curved GO surface so that its primary and secondary high potential regions were in close contact with the substrate with an equilibrium vertical distance of 3.031 Å from the oxygen in the epoxy group, while the negatively charged sites were left free from van der Waals contact. The adsorption energy of DA on the epoxidized GO surface was 0.044 Ry, slightly less than that of DA binding with the pristine graphene, but the binding stiffness was slightly increased to 0.155 Ry/Å^2^. For the UA molecule in Figure 3f, both the primary and secondary high potential regions were closely bound to four epoxy groups of the substrate, corresponding to the highest adsorption energy with the epoxidized GO. The UA molecule itself was slightly distorted by the curvature of the GO surface so that the distance between the ring center and the middle epoxy group became 2.919 Å, closer than the AA and DA cases. Although the energy penalty from the configurational stress should reduce the adsorption energy, the largest adsorption energy was found for UA binding on the epoxidized GO was with the largest adsorption energy of 0.048 Ry, equal to binding on the pristine graphene.

Figure 3g–i showed the optimized binding configuration of AA, UA, and DA on the G(OH)5-paraA substrate. The net partial charge of a hydroxyl functional group of GO was around −0.16 e, weaker than the net charge −0.37 e of an epoxy group. Therefore, the binding of AA and DA on the G(OH)5-paraA substrate was weaker than the GO5-para epoxidized substrate. For the case of AA binding on the G(OH)5-paraA substrate in Figure 3g, adsorption energy was decreased to 0.024 Ry due to weaker electrostatic interactions, despite the increase in van der Waals contacts due to the relatively planar surface of G(OH)5-paraA when compared to GO5-para. For the DA molecule binding with the G(OH)5-paraA substrate in Figure 3h, binding energy was significantly reduced to 0.026 Ry due to weaker electrostatic interactions. Steric clashes from the off-plane H7 and H8 atoms also contributed to the loss of adsorption energy, as the 2.859 Å distance from the closest hydroxyl H atom was the largest among all three analyte molecules. For the UA molecule on the G(OH)5-paraA substrate in Figure 3i, a smaller distance of 2.553 Å was measured from the closest hydroxyl H atom. The planar structure of UA corresponded to the absence of steric effects from off-plane atoms. As the C-O bonds of hydroxyl groups were rotatable, the direction of hydroxyl dipole moments was reoriented to maximize the absorption energy similarly with the electrostatic induction of the pristine graphene plane. As a result, the adsorption energy of the UA molecule on the G(OH)5-paraA substrate was equal to that of the UA molecule on the pristine graphene.

From Figure 3, the AA, DA, and UA analytes could be classified by the adsorption on the graphene functionalized by epoxy and hydroxyl groups. It could be seen that the adsorption energy of AA on the graphene plane was decreased when the graphene plane was functionalized by either epoxy and hydroxyl groups. Meanwhile, the adsorption of DA on the graphene plane decreased when the graphene plane was functionalized by epoxy groups but was unaffected by the hydroxyl functionalization, and the adsorption of UA was unaffected by both epoxy and hydroxyl groups.

### 3.4. AA, DA, and UA Charge Transfer Analysis

In this section, detailed mechanisms on how epoxy and hydroxyl functional groups affect the oxidation potentials of AA, UA, and DA molecules at GO and rGO surfaces will be discussed. As these analyte molecules were adsorbed, partial charge transfer occurred between the analyte and the substrate due to the overlapping of the off-plane molecular orbitals. Figure 4 displayed the optimized configurations of AA, DA, and UA analytes on the pristine graphene, GO5-para, and G(OH)5-paraA substrates with isosurfaces showing charge density difference (Δρ) [48] between the systems before and after substrate binding from
(4)$$\Delta \rho =\rho_{tot}-\rho_{substrate}-\rho_{analyte}$$
when $\rho_{tot}$ represented the charge density profile of the optimized binding configuration, $\rho_{substrate}$ represented the charge density profile of the unbound substrate, $\rho_{analyte}$ represented the charge density profile of the unbound analyte. The isosurfaces of charge density difference +0.001 e/Å^3^ (light blue color in Figure 4) represented the regions losing the probability to find electrons, while the isosurfaces of charge density difference −0.001 e/Å^3^ (yellow color in Figure 4) represented the regions gaining the probability to find electrons. Total charge transfer Δq for each system was also provided, in which the positive and negative signs of Δq denoted the loss and gain of probability to find electrons for the analyte molecules.

According to our potential energy calculations, the adsorption of AA, DA, and UA on the pristine graphene surface (Figure 4a–c) was stronger than the adsorption on the functionalized graphene due to the conductivity of graphene causing the regions of induced charge. Despite the strong adsorption, the larger intermolecular distance between analytes and pristine graphene corresponded to the less frequent pi orbital overlapping. Therefore, the smallest charge density and electronic transfer were observed for the pristine graphene than the functionalized substrates. For the case of AA, DA, and UA adsorption on the epoxidized GO5-para substrate (Figure 4d–f), the highly electronegative oxygen atoms tended to receive electrons from the pi-orbitals of analytes. Partial charge transfer occurred from the highly aromatic DA at +0.182 e and UA at +0.232 e to the GO5-para substrate, more than the charge transfer from AA at +0.047 e. The difference was due to the greater probability of pi-orbital overlapping from the aromaticity of DA and UA. For the hydroxyl-functionalized G(OH)5-paraA substrate binding of the analytes (Figure 4g–i), weaker interactions with the hydroxyl groups than the epoxy groups resulted in smaller electron transfer from the analytes. Only the UA molecule with the strongest adsorption energy on G(OH)5-paraA was with the positive Δq of the charge on the analyte molecules is given for each system.

Local partial charge transfer at each atom within an analyte molecule resulted in changes in charge distribution and local dipole moments. As a result, the partial charge around the oxidation sites was affected, which also affected the oxidation potentials. To further quantify this phenomena, partial charge differences of the analytes (i) in oxidized form, (ii) binding with pristine graphene, (iii) binding with GO5-para, and (iv) binding with G(OH)5-paraA were calculated compared to those of bare analytes for each ring member group and shown in Table 4, Table 5 and Table 6. Firstly, charge difference Δq(Ox) was calculated to address the changes between the neutral and oxidized forms of AA, DA, and UA. Partial charges at each of the oxidation sites were denoted by ‘*’ and were subtracted by 1 to represent the electronic state of the atom group just before losing the electron. For each analyte molecule, the sum of Δq(Ox) represented the net amount of local electron transfer from the neighboring atoms to the oxidation sites for an oxidation reaction. The partial charge analysis showed that the oxidation of a UA molecule required the highest amount of partial charge transfer at around −0.65 e, followed by a DA molecule at −0.37 e, and an AA molecule at −0.12 e. From this difference in Δq(Ox), it could be suggested that the largest electrostatic potential energy would be required for UA oxidation, corresponding to the highest oxidation potentials in CV and DPV experiments. It could also be seen that electron was transferred out from the atom groups adjacent to oxidation sites (see the underlined Δq(Ox) values in Table 4, Table 5 and Table 6).

Consider the partial charge difference between the atom groups in the analyte molecules before and after binding with the pristine graphene (Δq(G)). Slightly negative values of the Δq(G) for all analytes illustrated that the pristine graphene lost a relatively small number of electrons to the analytes, which might facilitate oxidation of all analytes with poor selectivity. Binding with the epoxidized GO5-para caused a more significant electron transfer from the analytes to the substrate than binding with the G(OH)5-paraA. Table 6 displayed five atom groups of UA with Δq> 0.025 e, including the C1/O3, C2, and C5/O2 groups adjacent to either H1/N1 and H2/N2 oxidation sites. The local electric fields created between the oxidation sites and the atom groups losing electrons prevented additional electron transfer to the oxidation sites. As greater potential energy was required to transfer charges to the oxidation sites of UA, a greater positive shift of the oxidation potentials for UA could be seen in the CV and DPV experiments. DA and AA were with three and one atom groups with 0.025 e or greater electron transfer, respectively. The smallest amount of electron transfer from AA to the G(OH)5-paraA substrate resulted in the lowest potential energy that was further required for oxidation. For the binding of analytes on the G(OH)5-paraA substrate, only two atom groups with electron loss of Δq(OH)> 0.025 e were found for UA, and none was found for the other two analytes, which rather received electron from the G(OH)5-paraA substrate.

## 4. Discussion

The information on the charge transfer between groups of atoms within the AA, DA, UA analytes, and the substrates provided insight into how the functionalization of graphene substrates could further differentiate the oxidation potentials of AA, DA, and UA during a simultaneous detection. According to the charge distribution analysis of the oxidized analyte molecules, electrons were transferred from neighboring atoms to the oxidation site prior to the oxidation reaction. The effects of physical adsorption on the pristine graphene, GO5-para, and G(OH)5-paraA substrates to the charge distribution within the analyte molecules might affect the oxidation. As an analyte molecule became adsorbed on the functionalized substrate, electronegativity of oxygen atoms in the functional groups caused a small amount of electron to transfer from the analyte to the substrate via pi-orbitals of the analyte. This effect was the most prominent for the epoxidized substrates with a relatively uniform negative partial charge of the functional groups and relatively high electronegativity. The spontaneous electron loss from the analyte to the substrate via this physical adsorption resulted in (i) an electric field between the analyte and the substrate that opposed further electron transfer from an oxidation reaction, and (ii) changes in charge distribution and local electric fields that prevented oxidation reactions. According to the partial charge difference analysis between the neutral analyte molecules and their oxidized forms, UA already required the largest local charge transfer to the oxidation site. The largest amount of electron loss from adsorption would only cause a further positive shift for the oxidation of UA. Meanwhile, the lower amount of electron loss from DA and AA adsorption resulted in a greater difference in oxidation potential, which was in agreement with experimental studies using the electrodes modified by rGO.

## 5. Conclusions

In this study, the mechanisms of oxidation potential shift of AA, DA, and DA molecules on the GO or rGO surfaces were illustrated through changes in the charge distribution within the analyte molecules at different oxidation and binding states. According to our DFT geometry optimizations, the strongest adsorption of UA to both types of functional groups corresponding to the largest amount of electron transfer through the pi orbitals. The local electric field created by the altered charge distribution within the analytes could prevent the oxidation and cause a further positive shift for the oxidation potential of UA from DA and AA. The more electronegative epoxy functional groups contributed to the charge transfer more than the hydroxyls. This understanding of this oxidation potential differentiation through the intrinsic properties of substrates and analyte molecules could be useful for other electrode modification designs of more reliable AA/DA/UA sensors by controlling the electron distribution of analytes via surface functionalization.

## Figures and Tables

**Figure 1 sensors-21-02773-f001:**
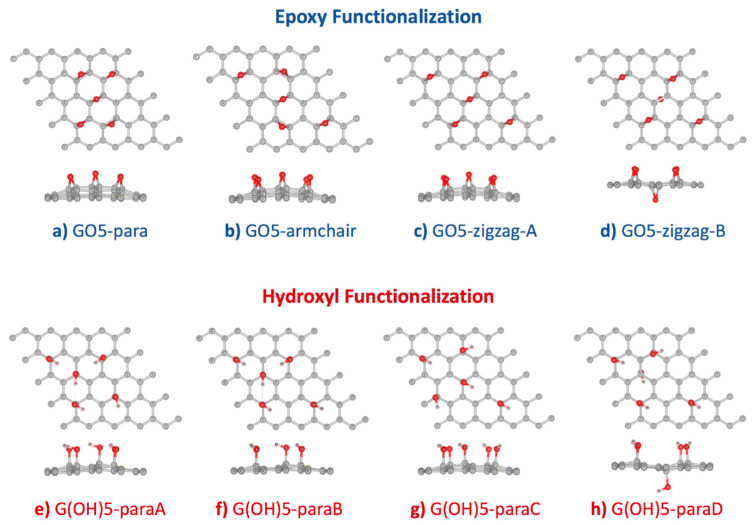
Optimized configurations of (**a**–**d**) 5 × 5 graphene supercell functionalized by five epoxy groups, and (**e**–**h**) 5 × 5 graphene supercell functionalized by five hydroxyl groups. Starting configurations followed Domancich et al. [42].

**Figure 2 sensors-21-02773-f002:**
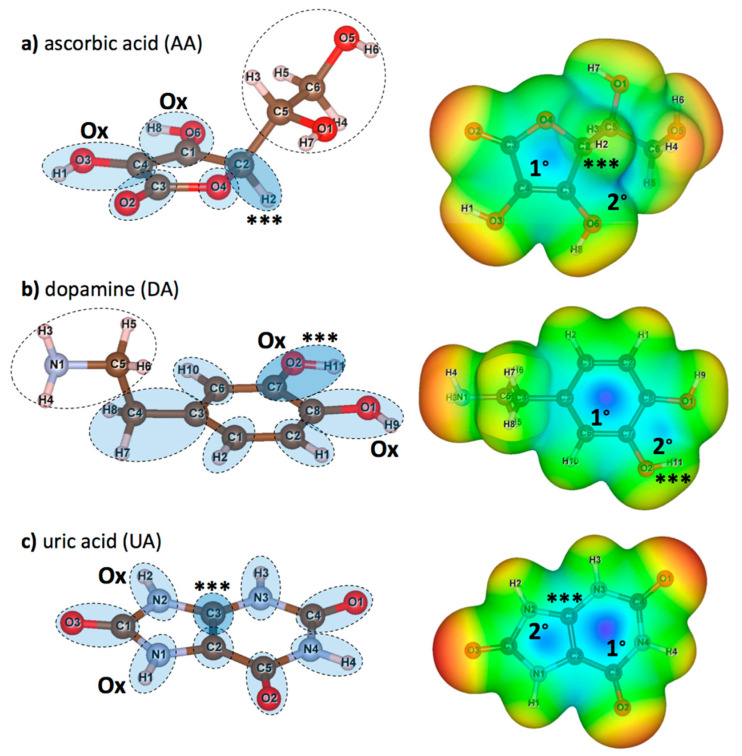
(left) atomic nomenclatures of neutral (**a**) ascorbic acid (AA), (**b**) dopamine (DA), and (**c**) uric acid (UA) molecules. Atom groups (dashed circles) were defined at each member of furanose and aromatic rings and its adjacent atoms. Additional groups of non-ringed atoms are also defined for AA and DA. Atom groups functioned as the sites for oxidation are labeled by ‘Ox’ and atom groups with highest positive partial charge (see Table 2) are labeled by ‘***’. (right) maps of relative electrostatic potential on the 0.005 e/Å^3^ iso-density surfaces of (**a**) AA, (**b**) DA, and (**c**) UA. Regions with relatively high potential are represented in blue, while regions with relatively low potential are represented in red. Primary (1°) and secondary (2°) high potential regions are also marked for further discussions.

**Figure 3 sensors-21-02773-f003:**
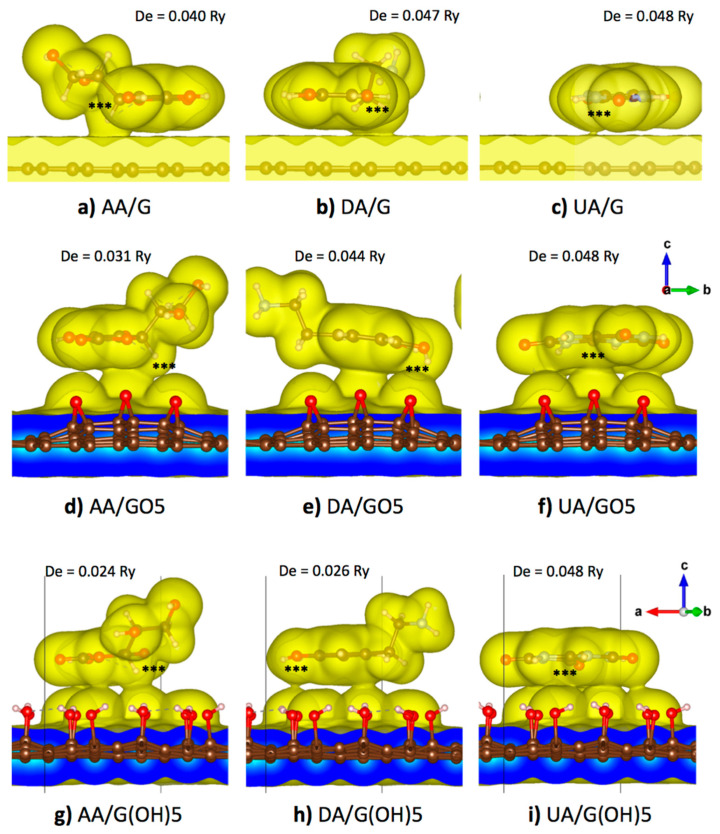
Optimized configurations and 0.005 e/Å^3^ iso-density surfaces of the (**a**) AA/graphene, (**b**) DA/graphene, (**c**) UA/graphene, (**d**) AA/GO5, (**e**) DA/GO5, (**f**) UA/GO5, (**g**) AA/G(OH)5, (**h**) DA/G(OH)5, and (**i**) UA/G(OH)5 systems. Atom groups with highest positive partial charge are labeled by ‘***’ and the adsorption energy from the Morse potential fitting is given for each system.

**Figure 4 sensors-21-02773-f004:**
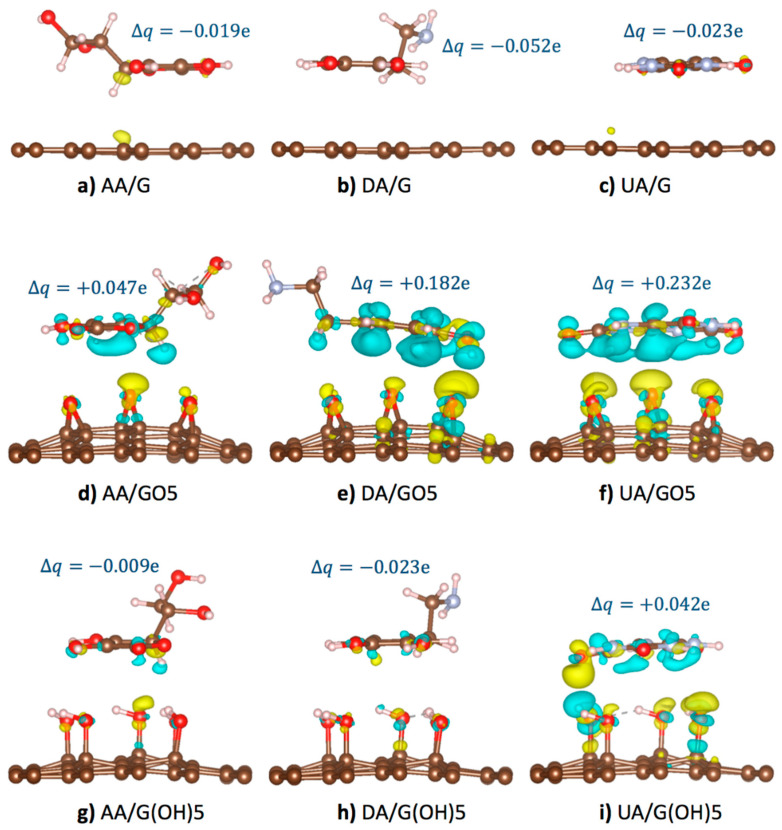
Optimized binding configurations, +0.001 e/Å^3^ isosurfaces of charge density difference after substrate binding (light blue), and −0.001 e/Å^3^ isosurfaces of charge density difference after substrate binding (yellow) of the (**a**) AA/graphene, (**b**) DA/graphene, (**c**) UA/graphene, (**d**) AA/GO5, (**e**) DA/GO5, (**f**) UA/GO5, (**g**) AA/G(OH)5, (**h**) DA/G(OH)5, and (**i**) UA/G(OH)5 systems. Total change Δq of the charge on the analyte molecules is given for each system.

**Table 1 sensors-21-02773-t001:** Formation energy of five epoxy or hydroxyl groups on the 5 × 5 graphene supercell under configuration proposed by Domancich et al.

System	Epoxy Functionalization		Hydroxyl Functionalization	
	*E_form/normal_* (eV)	∆*z* (Å)	*E_form/normal_* (eV)	∆*z* (Å)
GO5-para	−3.195	1.126	−1.440	0.897
GO5-armchair	−3.092	1.046	−1.348	0.751
GO5-zigzag-A	−3.140	0.998	−1.191	0.894
GO5-zigzag-B	−3.231	0.555	−1.259	0.621

**Table 2 sensors-21-02773-t002:** Partial charge on each atom of AA, DA, and UA calculated from the Löwdin population of valence electrons. Total charge on each of the atom groups defined in Figure 2 is also shown. The bold fonts highlight the atom group with the highest positive partial charge for each analyte.

AA			DA			UA		
*furanose*			*aromatic*			*aromatic*		
C1	0.186	0.106	C1	−0.176	−0.034	C1	0.513	0.032
O6	−0.476		H2	0.142		O3	−0.481	
H8	0.397							
			C2	−0.200	−0.055	H1	0.324	−0.025
C4	0.085	−0.029	H1	0.145		N1	−0.349	
O3	−0.508							
H1	0.395		C8	0.184	0.029	C2	−0.018	−0.018
			H9	0.373				
C3	0.502	0.061	O1	−0.528		**C3**	**0.219**	**0.219**
O2	−0.440							
			**C7**	**0.227**	**0.098**	H2	0.312	−0.055
O4	−0.341	−0.341	**H11**	**0.378**		N2	−0.367	
			**O2**	**−0.507**				
**C2**	**0.040**	**0.211**				C5	0.412	−0.069
**H2**	**0.171**		C6	−0.189	−0.033	O2	−0.481	
			H10	0.156				
*non−furanose*						H4	0.317	−0.076
			C3	0.016	0.007	N4	−0.393	
C5	0.053	−0.009	C4	−0.281				
C6	−0.091		H7	0.128		C4	0.531	0.043
O1	−0.579		H8	0.142		O1	−0.488	
O5	−0.573							
H3	0.136		*non−aromatic*			H3	0.314	−0.047
H4	0.131					N3	−0.361	
H5	0.171		C5	−0.189	−0.061			
H6	0.366		H3	0.267				
H7	0.375		H4	0.258				
			H5	0.142				
			H6	0.110				
			N1	−0.650				

**Table 3 sensors-21-02773-t003:** Morse potential fitting parameters of AA, DA, and UA adsorption obtained from the calculated potential energy as functions of distances from the pristine graphene, GO5 and G(OH)5 substrates.

**System**	***De* (Ry)**	***α* (1/Å)**	***r_e_* (Å)**	***k_e_* (Ry/Å^2^)**
AA/Graphene	0.040	1.296	3.439	0.134
DA/Graphene	0.047	1.230	3.573	0.142
UA/Graphene	0.048	1.320	3.424	0.167
AA/GO5	0.031	1.339	3.052	0.111
DA/GO5	0.044	1.326	3.031	0.155
UA/GO5	0.048	1.350	2.919	0.175
AA/G(OH)5	0.024	1.427	2.720	0.098
DA/G(OH)5	0.026	1.369	2.859	0.097
UA/G(OH)5	0.048	1.379	2.553	0.183

**Table 4 sensors-21-02773-t004:** Change in partial charge ∆*q* of each atom group of an ascorbic acid (AA) from the neutral state to the oxidized state (∆*q*(Ox)), and the bound state with the pristine graphene (∆*q*(G)), GO5-para (∆*q*(GO5)), and G(OH)5-paraA (∆*q*(OH)). For the oxidized molecule (italic), the ∆*q* of oxidized sites (bold) and their adjacent atom groups (underlined) are highlighted. For the bound molecules, the ∆*q* of atom groups losing more than 0.025 electrons to the substrate via adsorption (bold) are highlighted. Oxidation sites are labeled by ‘*’.

AA	∆*q*(Ox)	∆*q*(G)	∆*q*(GO5)	∆*q*(OH)
C1/O6/H8 *	***−0.128***	−0.001	0.017	−0.014
C4/O3/H1 *	***0.011***	−0.003	0.021	−0.005
C3/O2	*0.079*	0.007	0.015	0.032
O4	*0.013*	−0.030	−0.030	−0.029
C2/H2	*−0.037*	0.008	**0.025**	0.007
***Total***		***−0.019***	***0.047***	***−0.009***

**Table 5 sensors-21-02773-t005:** Change in partial charge ∆*q* of each atom group of a dopamine (AA) from the neutral state to the oxidized state (∆*q*(Ox)), and the bound state with the pristine graphene (∆*q*(G)), GO5-para (∆*q*(GO5)), and G(OH)5-paraA (∆*q*(OH)). For the oxidized molecule (italic), the ∆*q* of oxidized sites (bold) and their adjacent atom groups (underlined) are highlighted. For the bound molecules, the ∆*q* of atom groups losing more than 0.025 electrons to the substrate via adsorption (bold) are highlighted. Oxidation sites are labeled by ‘*’.

DA	∆*q*(Ox)	∆*q*(G)	∆*q*(GO5)	∆*q*(OH)
C1/H2	*0.036*	−0.004	**0.029**	−0.009
C2/H1	*0.045*	0.003	**0.038**	−0.008
C8/O1/H9 *	***−0.200***	−0.002	**0.062**	0.009
C7/O2/H11 *	***−0.173***	−0.018	0.009	0.001
C6/H10	*−0.009*	−0.008	0.007	−0.005
C3/C4/H7/H8	*0.024*	−0.023	0.037	−0.011
***Total***		***−0.052***	***0.182***	***−0.023***

**Table 6 sensors-21-02773-t006:** Change in partial charge ∆*q* of each atom group of a uric acid (UA) from the neutral state to the oxidized state (∆*q*(Ox)), and the bound state with the pristine graphene (∆*q*(G)), GO5-para (∆*q*(GO5)), and G(OH)5-paraA (∆*q*(OH)). For the oxidized molecule (italic), the ∆*q* of oxidized sites (bold) and their adjacent atom groups (underlined) are highlighted. For the bound molecules, the ∆*q* of atom groups losing more than 0.025 electrons to the substrate via adsorption (bold) are highlighted. Oxidation sites are labeled by ‘*’.

UA	∆*q*(Ox)	∆*q*(G)	∆*q*(GO5)	∆*q*(OH)
C1/O3	*0.207*	−0.004	**0.052**	0.010
H1/N1 *	***−0.298***	−0.023	0.000	−0.026
C2	*0.080*	0.001	**0.036**	−0.005
C3	*0.040*	0.022	**0.049**	**0.026**
H2/N2 *	***−0.354***	−0.009	−0.001	−0.006
C5/O2	*0.205*	0.005	**0.052**	0.014
H4/N4	*0.026*	−0.012	0.001	−0.008
C4/O1	*0.072*	0.005	**0.034**	**0.032**
H3/N3	*0.055*	−0.009	0.009	0.005
***Total***		***−0.023***	***0.232***	***0.042***

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
