# Peer review of "Unveiling the Fundamental Mechanisms of Graphene Oxide Selectivity on the Ascorbic Acid, Dopamine, and Uric Acid by Density Functional Theory Calculations and Charge Population Analysis"

_sensors, 2021, doi:10.3390/s21082773_

Round 1
Reviewer 1 Report
This theoretical manuscript discusses the application of a DFT formalism to the analysis of adsorption of ascorbic acid (AA), dopamine (DA) and uric acid (UA) to pristine and functionalized graphene interfaces. Optimization of geometries and energies of the three analytes, as well as analysis of the charge transfer between AA, DA and UA and the graphene supports is presented and clearly discussed.
In my opinion, this is a nice contribution to this type of research. No relevant drawbacks appear and the different aspects of the manuscript are analysed in detail. Only two minor formal questions could be revised by the authors.
- First, there is a broad literature relative to the problems arising when simultaneous detection of AA, DA and UA is done, as the authors indicated in page 1, lines 40, 41. Some relevant reviews could be included in this line.
- Columns Eform and Eform/normal in Table 1 are redundant. The authors should include only one.
I consider that, once this minor points were corrected, this manuscript is suitable for being accepted in Sensors.
Author Response
Comment 1: First, there is a broad literature relative to the problems arising when simultaneous detection of AA, DA and UA is done, as the authors indicated in page 1, lines 40, 41. Some relevant reviews could be included in this line.
Response: A review has been cited in line 42. DA detections under the interference by AA and UA were mentioned.
Comment 2: Columns Eform and Eform/normal in Table 1 are redundant. The authors should include only one.
Response: The Table 1 has been edited and highlighted in the new manuscript to remove the redundant content. Thank you very much.
Reviewer 2 Report
The manuscript described modelling of electrochemical process of sensors based on GO and rGO for ascorbic acid, dopamine and uric acid. The mechanisms of oxidation potential shift of analyte molecules on these surfaces were described and calculated through changes in the charge distribution within the molecules at different oxidation and binding states. The manuscript is clearly written and helps electrochemists to predict sensor properties. The quality of the manuscript is excellent and it can be published as it is.
Author Response
All authors thank the reviewer for his/her time and comments.
Reviewer 3 Report
The selectivity of electrochemical sensors to ascorbic acid (AA), dopamine (DA), and uric acid (UA) remains an open challenge in the field of biosensing. In this study, the selective mechanisms for detecting AA, DA, and UA molecules on the graphene and graphene oxide substrates were illustrated through the charge population analysis from the DFT calculation results. From the calculations, AA was bound to both epoxy and hydroxyl-functionalized GO with relatively low adsorption energy, while DA was adsorbed stronger to the electronegative epoxy groups. The strongest adsorption of UA to both types of functional groups corresponded to the largest amount of electron transfer through the pi orbitals of UA. This work has great significance for design biosensors for simultaneous detection of AA, DA and UA. So, I recommend accept the manuscript after minor revision. This manuscript has minor spelling typos, style errors and grammatical errors, please correct them. In the introduction, the recent voltametric sensors should be cited, such as Materials Science & Engineering C, 2020,109, 110615; Microchimica Acta, 2020, 187(2): 1-10; Sensors and Actuators B: Chemical, 2020, 314: 128059. New Journal of Chemistry, 2020, 44(12): 4916-4926.
Author Response
Comment 1: This manuscript has minor spelling typos, style errors and grammatical errors, please correct them.
Response: A revision was done to address the spelling typos and grammatical errors. Changes were highlighted in blue color.
Comment 2: In the introduction, the recent voltametric sensors should be cited, such as Materials Science & Engineering C, 2020,109, 110615; Microchimica Acta, 2020, 187(2): 1-10; Sensors and Actuators B: Chemical, 2020, 314: 128059. New Journal of Chemistry, 2020, 44(12): 4916-4926.
Response: All these references have been added to the lines 56-58 of the revised manuscript. Thank you very much.
Reviewer 4 Report
The authors describe the density functional theory calculations and charge population analysis for reveals the fundamental mechanisms of graphene oxide selectivity on the AA, DA, and UA. it is a good work where the authors have discovered a fundamental mechanism of GO selectivity on electrochemical detection of AA, DA, and UA. In my opinion, the paper is suitable for publication in Sensors.
Author Response

(The authors gave the same response as above.)
